# Reduced Graphene Oxide-Zinc Sulfide Nanocomposite Decorated with Silver Nanoparticles for Wastewater Treatment by Adsorption, Photocatalysis and Antimicrobial Action

**DOI:** 10.3390/molecules28030926

**Published:** 2023-01-17

**Authors:** Hina Naeem, Hafiz Muhammad Tofil, Mohamed Soliman, Abdul Hai, Syeda Huma H. Zaidi, Nadeem Kizilbash, Daliyah Alruwaili, Muhammad Ajmal, Muhammad Siddiq

**Affiliations:** 1Department of Chemistry, Rawalpindi Women University, 6th Road, Satellite Town, Rawalpindi 46300, Pakistan; 2Department of Chemistry, Quaid-i-Azam University, Islamabad 45320, Pakistan; 3Department of Microbiology, Faculty of Medicine, Northern Border University, Arar 91431, Saudi Arabia; 4Department of Medical Laboratory Technology, Faculty of Applied Medical Sciences, Northern Border University, Arar 91431, Saudi Arabia; 5Department of Chemistry, Faculty of Science, Northern Border University, Arar 91431, Saudi Arabia; 6Department of Chemistry, Division of Science and Technology, University of Education, Lahore 54770, Pakistan

**Keywords:** graphene, silver, nanocomposite, adsorption, photocatalysis, antibacterial

## Abstract

Reduced graphene oxide nanosheets decorated with ZnS and ZnS-Ag nanoparticles are successfully prepared via a facile one-step chemical approach consisting of reducing the metal precursors on a rGO surface. Prepared rGO-ZnS nanocomposite is employed as an adsorbent material against two model dyes: malachite green (MG) and ethyl violet (EV). The adsorptive behavior of the nanocomposite was tuned by monitoring some parameters, such as the time of contact between the dye and the adsorbent, and the adsorbent dose. Experimental data were also simulated with kinetic models to evaluate the adsorption behavior, and the results confirmed that the adsorption of both dyes followed a pseudo 2nd order kinetic mode. Moreover, the adsorbent was also regenerated in a suitable media for both dyes (HCl for MG and ethanol for EV), without any significant loss in removal efficiency. Ag doped rGO-ZnS nanocomposite was also utilized as a photocatalyst for the degradation of the selected organic contaminant, resorcinol. The complete degradation of the phenolic compound was achieved after 60 min with 200 mg of rGO-ZnS-Ag nanocomposite under natural sunlight irradiation. The photocatalytic activity was studied considering some parameters, such as the initial phenol concentration, the photocatalyst loading, and the pH of the solution. The degradation kinetics of resorcinol was carefully studied and found to follow a linear Langmuir–Hinshelwood model. An additional advantage of rGO-ZnS and rGO-ZnS-Ag nanocomposites was antibacterial activity against Gram-negative bacterium, *E. coli*, and the results confirmed the significant performance of the nanocomposites in destroying harmful pathogens.

## 1. Introduction

Over the last few years, the use of phenolic compounds and their derivatives has increased a great deal in diverse industries, such as the paper, pulp, dyes, pharmaceuticals, pesticides, and food industries [1]. They have become a serious threat to human beings, the atmosphere, and above all, aquatic life due to their carcinogenic nature and their low biodegradability. The US Environmental Protection Agency (EPA) has listed phenolic compounds as a primary, as well as a priority, threat to aquatic life due to their large-scale release in the environment annually. Apart from this, various colored organic pollutants, such as dyes, are also causing harm to humans and aquatic life, as they are also the main effluent of an extendable range of industries [2]. These toxic phenolic compounds and dyes are becoming persistent pollutants of the modern era due to their higher chemical stability, and their removal from waste water is an utmost need of society to improve the quality of consumable water [3].

Various studies to establish appropriate methods for the mineralization of these pollutants have been reported previously. These methods include ion exchange, reverse osmosis, adsorption, and electrochemical and biological treatment [1,4,5]. However, some of these methods incur higher costs and extended time duration and are also responsible for the generation of secondary pollutants [6]. Recently, adsorption is considered to be a more appropriate method due to its simple procedure, easy operational method, and suitability for various dyes. Further, no degradation of pollutants is observed in the adsorption process, as degradation products of some dyes are also harmful and after adsorption, adsorbent can be easily removed from the water.

Various adsorbents have been developed to find the economical and handy adsorbents for the adsorptive removal of unwanted contaminants from water. A lot of research has been focused on carbon-based adsorbents. For example, granular activated carbon adsorbents have been widely explored for water treatment. Granular activated carbon can be obtained from the waste materials using different methods, depending upon the nature of waste material, as has been already reviewed by Jjagwe et al. [7]. The development of carbon-based adsorbents from the waste material for the treatment of contaminated water has emerged as a fascinating way to improve the economy, handling, and efficiency of the water treatment process. For example, Obey et al. [8] recently reported that biochar can be prepared from an agro-waste matamba fruit shell and can be employed as the adsorbate for the removal of iodine. In this study, authors found that the composition of the obtained biochar provides enough polarization ability, which in turn causes the high iodine adsorption that was significantly induced by weak van der Waals forces and π-π and π-stacking interaction on the biochar surface and its micropores. The carbon-based reusable magnetic WC-Co adsorbents for the removal of dyes from contaminated water have also been reported by Xiaofan et al. [9].

In addition to the carbon-based adsorbents, many other materials have also been explored as adsorbents for the removal of various water contaminants. For example, Naseem et al. [10] have reported the adsorptive removal of heavy metal ions from water by a hydrogel. Shehzad et al. [11] employed modified alginate-chitosan-TiO_2_ composites as adsorbents for the removal of toxic metal ions from water. Naseem et al. [12] used husk biomass as the adsorbent for the removal of water contaminants.

Many researchers reported nanocomposites as efficient adsorbent materials, in which inorganic particles were synthesized onto the graphene sheets through electrostatic interactions, and then metal ions were reduced into metal nanoparticles [13]. This type of hybridization not only enhances the adsorption capacity of nanocomposites, but also prevents restacking of graphene layers and provides additional advantages of catalytic and antimicrobial activities for the treatment of multiple water pollutants. Cheng et al. [14] fabricated magnetic composite gels by combining GO with poly (vinyl alcohol) and platinum (Pt) nanoparticles and studied their adsorptive and catalytic behavior. The authors reported the higher adsorption capacity and simple separation capability of prepared composite gels by applying an external magnetic field. Moreover, composite gels comprised of Pt nanoparticles were found to have excellent catalytic properties. Similarly, Wang et al. [15] reported a reduced graphene oxide-based nanocomposite decorated with polypyrrole and iron oxide nanoparticles. The prepared nanocomposite showed a high adsorption capacity for chromium (VI). Yuan et al. [16] prepared nanocomposite by decorating silver nanoparticles on reduced GO and found it to be a stable photocatalyst with reasonably high catalytic efficiency. However, for the removal of aromatic pollutants from aqueous medium, adsorption was not found to be a favorable process.

The advanced oxidative process (AOP), more specifically photocatalysis, is considered to be a more versatile process that showed significant improvement in producing pollutant-free water against aromatic pollutants. Photocatalytic remediation of pollutants using nanostructured heterogenous photocatalysts under a UV or visible light source has increased tremendously over the last few years due to low cost, easy synthesis, and feasible handling [17,18]. In photocatalytic degradation, the catalyst is irradiated with photons of energy equal to or greater than its band gap, which excites an electron from the valence band to the conduction band, producing a hole in the valence band. This electron-hole pair produces reactive oxygen species (ROS) by interacting with the aqueous medium. These ROS are involved in degrading pollutants on the photocatalyst surface. A large number of semiconducting nanomaterials, such as WO_3_, ZnO, TiO_2_, and Fe_3_O_4_, were utilized as photocatalysts previously [19]. Apart from these conventional photocatalysts, ZnS has gained popularity as an efficient photocatalyst due to its low cost, non-toxicity, abundance, and photostability [20,21,22]. However, there are still some concerns that need to be resolved for its use on an industrial scale, such as a higher band gap and its agglomerating tendency during the synthesis process, along with its difficult removal from the reaction medium [22,23,24]. Although many materials and methods, as mentioned above, have been designed and established for the treatment of contaminated water, they can be used for the treatment of one type of pollutant by either adsorption, catalysis, or antimicrobial action. For the advancement of technology, there is a need to design a multifunctional system that can be used to treat multiple pollutants with different actions, such as adsorption, catalysis, or antimicrobial action. To this end, we have designed a material that not only can act as adsorbent, but also shows photocatalytic and antimicrobial properties for the removal of aromatic pollutants and microbes from water.

In the first step, rGO was decorated with ZnS to design a rGO-ZnS composite to be used as the adsorbent material for the removal of the dyes malachite green (MG) and ethyl violet (EV). In the second step, doping of rGO-ZnS with low-cost noble metal, such as silver nanoparticles (NPs), was carried out to design a rGO-ZnS-Ag nanocomposite with photocatalytic and antimicrobial activities. This doped nanomaterial showed much venerated photocatalytic behavior in the present work.

The prepared rGO-ZnS and rGO-ZnS-Ag nanocomposites were characterized in detail using different spectroscopic, diffraction, and microscopic techniques and showed much improved photocatalysis, adsorption, stability, and recyclability, as compared to previously reported materials [24,25]. Along with this, the influence of various other factors on photocatalysis, such as the pH, photocatalyst amount, concentration of pollutants, and adsorbent dose in waste water, were also studied in detail in order to optimize the adsorptive and photocatalytic behavior of the prepared nanocomposites. Moreover, synthesized rGO-ZnS nanocomposite also showed excellent stability for five consecutive cycles of adsorption-desorption.

## 2. Results and Discussion

### 2.1. Characterization

The XRD pattern in Figure 1 shows the crystalline structure and phase of the rGO, rGO-, and rGO-ZnS-Ag nanocomposite. Sharp diffraction peaks with no extra signal confirmed the formation of crystalline phase nanocomposite material [26]. For ZnS, peaks positions 28.7°, 48.2°, and 57.0° correspond to hkl values of (111), (220), and (311), showing cubic crystalline phase. The Debye Scherrer formula calculated average crystallite size of 2.87 nm for ZnS in the composite. In the case of rGO-ZnS and rGO-Ag-ZnS nanocomposites, no characteristic signal for rGO is present in the XRD pattern, possibly due to the growth of zinc sulfide nanoparticles between rGO layers, hence destroying its regular layered structure [27].

The TEM images of the rGO-ZnS (Figure 2a,b) sheet-like morphology of the reduced graphene with prominent elevations and depressions can be seen in the TEM images of the synthesized nanocomposites. The ZnS nanoparticles over the reduced graphene surface are somewhat spherical in shape, with varying size. In addition, the nanoparticles are uniformly distributed over the graphene substrate, with an average particle size of 4.91 nm with a lesser degree of agglomeration [28]. The TEM image of rGO is given in Appendix A, which shows a wavey and wrinkle-like pattern. The TEM image of rGO-ZnS-Ag is given in Appendix A, which shows the presence of Ag-doped ZnS on the rGO with block dots.

A quantitative analysis by EDX, along with elemental mapping, was performed for the rGO-ZnS and rGO-ZnS-Ag nanocomposites in order to have a look at the composition of the nanocomposite. As shown in Figure 3a, the prepared nanocomposites are composed of C, O, Zn, S, and Ag elements. Furthermore, as shown in Figure 3b, SEM-EDX elemental mapping confirmed that the elements are not confined in a limited space; rather, they are distributed throughout the rGO matrix [29]. The existence of C and O peaks in the EDX spectrum also reinforced the graphene presence in the nanocomposites. No extra signal is present in the EDX spectrum, confirming the purity of the samples.

FT-IR analysis of the rGO-ZnS-Ag, rGO-ZnS, and rGO in Figure 4a–c, respectively, display some characteristic peaks of the graphene oxide, i.e., 1727 cm^−1^ for the C=O stretching vibration of the carboxyl moiety and 1218 cm^−1^ for the C-OH stretching [30,31]. The peaks corresponding to the oxygen functionalities of GO, such as carbonyl, hydroxyl, carboxyl, and epoxy groups, disappeared completely upon reduction, whereas some other peaks underwent a substantial shift from 1048 cm^−1^ to 1005 cm^−1^ for the C-O stretching vibration due to the strong reduction of the graphene oxide by the N, N Dimethyl formamide, indicating the good reduction ability of the solvent [22]. Thus, DMF not only acts as a medium for nanocomposite formation, but also a reducing agent for both nanoparticles and for GO sheets. Furthermore, a new band appears at 876 cm^−1^ due to the bending mode of the aromatic C-H groups on the reduced graphene surface, along with the appearance of the characteristic bands of the ZnS and Ag at 1116 and 668 cm^−1^, supported by the formation of nanoparticles over the graphene substrate [32]. A broad absorption band around 3400 cm^−1^ was observed in all the three samples, which is associated with the O-H stretching vibration of the C-OH groups and the water entrapped in the structures. The absorption peaks at 1644 cm^−1^ appeared due to the C=C stretching [33].

### 2.2. Nitrogen Gas Adsorption/Desorption Analysis

To determine the surface area, pore radius, and pore volume of the prepared rGO-ZnS-Ag nanocomposite, nitrogen gas adsorption/desorption analysis was performed. The corresponding BET and Langmuir isotherms are shown in Appendix A, respectively. The surface area was determined via the single-point and multi-point BET, Langmuir, DFT, BJH, and DH methods. In addition to the surface area, the pore volume and pore size were also determined, as shown in the Table 1. The values of the above-mentioned parameters determined from the different methods were close to each other, except for the surface area determined from the Langmuir and DFT methods. The BET surface areas 456.1 and 550.3 m^2^/g determined from the single-point and multi-point methods, respectively, was larger as compared to the previous reports [34,35,36].

### 2.3. Adsorption Test

The adsorption test of rGO-ZnS was conducted on the removal of the two model dyes MG and EV from the aquatic medium in the present work. Their adsorbed amounts onto the rGO-ZnS nanocomposite were calculated in terms of their absorbance values at λmax = 617 nm and 595 nm for MG and EV, respectively. According to previous literature reports, reduced graphene nanocomposites with metal sulfides have not been employed in environmental applications as an adsorbent material. Rather, their oxides, such as zinc oxide (ZnO), magnesium oxide (MgO), and cadmium oxide (CdO), etc., have been involved in various ways as a photocatalyst for dyes and other aromatic pollutants degradation [37]. Yet, no such report has been cited for their use as an effective adsorbent material against various pollutants. Thus, for the first time, the effectiveness of the prepared rGO-ZnS nanocomposite in the removal of dyes from an aqueous environment was studied in detail in the present work. Generally, the adsorption process is highly influenced by various experimental parameters, such as the adsorbent amount, the dyes initial concentration, pH, and the temperature of the reaction medium. Therefore, the effects of these parameters were studied thoroughly.

### 2.4. Effect of Dyes Initial Concentration on Adsorption Rate

The effectiveness of the prepared rGO-ZnS nanocomposite in the dyes removal from the aqueous environment was studied with four different starting concentrations (3, 5, 7, and 10 µM) of the MG and EV dyes, and the results are given in Figure 5a,b, respectively. According to the these, most of the initial concentrations of the dyes were removed from the aqueous medium within the first 35 min. Moreover, an increase in the dyes’ initial concentration increases the adsorption rates and the adsorbed amounts of the dyes. Therefore, a concentration gradient was developed between the dyes in the bulk solution and the dyes’ concentration on the adsorbent surface [38].

With the increase in the dyes’ initial concentration, larger numbers of molecules approach and adsorb at the surface of the adsorbent. Figure 5a,b also evidenced that the removal rate was rapid during the first 20 min and later becomes slow. This rapid initial adsorption process is due to the greater number of available active sites on the nanocomposite [38]. An adsorption capacity is practically an important parameter in the industrial application of an adsorbent, so it was evaluated that the equilibrium values were found to increase from 17.3 mg/g to 37.99 mg/g in the case of MG and from 11.95 mg/g to 23.95 mg/g in the case of EV, with an increase in starting concentration from 3 µM to 10 µM. The adsorption equilibrium was established within 10–15 min, which is far better than many previous studies, where it was achieved after a longer time [12,39]. These values indicate that the prepared system can be used as an efficient adsorbent for the removal of various organic pollutants.

Three different amounts of adsorbent i.e., 3, 5, and 7 mg, were used to study the effect of the adsorbent dose on the rate of dyes removal from the aqueous medium, keeping all other parameters constant, e.g., the concentrations of MG and EV (5 µM), the volume of the solutions (50 mL), and the temperature (25 °C). Experimental results revealed that at the fixed dyes concentrations, the equilibrium adsorbed amount increases from 30 mg/g to 60 mg/g in the case of MG and from 20 mg/g to 35.78 mg/g in the case of EV dye, with the increase in the nanocomposite dosage, as shown in Appendix A and b, respectively. These results are consistent with the availability of the greater number of active sites in the adsorption medium and, hence, the greater interaction with the dye molecules. A similar increase in the adsorbed amounts and adsorption rates has already been observed and reported [40]. The adsorption observed in this was found to be much better, as compared to the adsorbents previously reported in the literature [41,42].

### 2.5. Adsorption Kinetics

Two very well-known kinetic models, pseudo 1st order and pseudo 2nd order rate equations, were studied for the rGO-ZnS nanocomposite with an initial concentration of 7 µM for both dyes.

### 2.6. Pseudo 1st Order Kinetics

The experimental data under study were treated with the pseudo first order kinetic model. According to the kinetic equation, log (q_e_ − q_t_) is plotted against time t, which is shown in Figure 5c. The k_1_ and q_e_ values are calculated from the slope and intercept and are given in Table 2.

### 2.7. Pseudo 2nd Order Kinetics

Graphs for the pseudo second order kinetic equation were obtained by plotting t/q_t_ as a function of time and are shown in Figure 5d for both dyes. From Table 2, it can be seen that R^2^ for the straight line graphs of both dyes for the pseudo 2nd order reaction are found to be nearly 1, as compared to that observed for the pseudo 1st order reaction. Additionally, the values of q_e_ calculated from the pseudo second order equation were found to be 27.54 mg/g and 20.04 mg/g for MG and EV, respectively, which matches well with the experimental values of 25.72 mg/g and 18.34 mg/g for both dyes. Thus, the results concluded that both the dyes and the prepared nanocomposite are readily involved in the adsorption process, and all the adsorption sites were efficient in adsorbing dyes molecules. This is also closely related with the previously stated results for the adsorption of various dyes on rGO sheets [43,44].

### 2.8. Reusability of Adsorbent

Considering the cost effectiveness and practical application of the prepared material on the industrial scale, reusability was studied by considering the desorption of both dyes in 0.1 M HCl and ethanol solvent for MG and EV dyes, respectively. Almost 85% of MG was desorbed in 0.1 M HCl and 10% in ethanol. Similarly, almost 7% of EV was desorbed in 0.1 M HCl and 80% in ethanol. After this, the nanocomposite was washed thoroughly with distilled water after its first desorption cycle and then used again for the adsorption of dyes, and the process was repeated for four consecutive cycles. Figure 6a,b shows the reusability of the adsorbent for both the MG and EV dyes, respectively.

According to these results, a negligible loss of 10% in adsorption capacity was noted until four consecutive cycles of adsorption-desorption. Thus, the combined advantages of easy separation from the adsorption medium, good desorption efficiency, and reusability suggests that rGO-ZnS can be used as an active and economical adsorbent for waste water treatment [45].

### 2.9. Photocatalytic Degradation of Resorcinol

The photocatalytic performance of synthesized rGO-ZnS-Ag nanocomposites was investigated for resorcinol degradation as the model organic pollutant under natural sunlight irradiation. The photodegradation of resorcinol using rGO-ZnS-Ag nanocomposite is measured using a UV-visible spectrophotometer at a wavelength of 274 nm. Without any catalyst, a very small change in the UV-visible spectrum can be observed in Appendix A. In the presence of photocatalyst, the reaction mixture was kept in the dark for 1 h to establish an adsorption-desorption equilibrium before exposure to light. As shown in Appendix A, a slight decrease of 0.069 in the absorption was observed, which can be associated with the adsorption of some amount of resorcinol on the surface of the photocatalyst. However, about 90% resorcinol removal from the aqueous environment was attained after 150 min at pH 7 in the presence of rGO-ZnS-Ag nanocomposite. Photocatalytic degradation depends on various influential factors, such as initial solution pH values, initial pollutant concentration, and catalyst amount. Thus, the influence of these parameters on the degradation of resorcinol in the photocatalytic reaction was investigated in detail.

### 2.10. Effect of Initial pH

Initial solution pH is one of the significant factors affecting photocatalytic removal of pollutants from the aqueous environment due to its effect on the catalyst surface charge and adsorption properties of the material [46]. Keeping the initial concentration of resorcinol solution constant at 0.5 mM, the catalyst concentration at 50 mg, and the irradiation time at 150 min, the initial pH values varied from 2 to 10. As can be seen in Figure 7, the highest degradation efficiency was observed at a pH value of 7, which means that a neutral environment is more suitable for resorcinol removal than an alkaline and acidic environment. In a very strong acidic and alkaline medium, the oxygen functionalities on rGO-ZnS-Ag nanocomposite tends to dissolve quickly, making the nanocomposite unstable. Thus, the prepared catalyst tends to lose its activity, and the efficiency of the degradation process decreases [47,48].

### 2.11. Effect of Catalyst Dose on Photodegradation

Another important parameter in monitoring the degradation behavior of resorcinol is the effect of the catalyst dose on the photocatalytic removal of the pollutant at its concentration (0.5 mM and pH 7) under direct sunlight irradiation. Four different catalyst amounts of 50, 100, 150, and 200 mg are used in the present study. Appendix A displays the effect of the catalyst on the degradation of the resorcinol in the aqueous medium. It was observed that the rate of time to complete the reaction was gradually decreased from 6 h to 20 min by a corresponding increase in the amount of the catalyst from 50 to 200 mg. This observation was obvious to the increase in the number of catalytically active sites available for the reactants, and hence, an increase in the collision frequency of the reactants, which in turn increases the rate of the reaction and decreases the time for the completion of the reaction.

### 2.12. Degradation Kinetics

Degradation kinetics study was carried out at the 0.5 mM resorcinol concentration and at a fixed pH value of 7, using different catalyst dosages of 50, 100, 150, and 200 mg. Because of the low resorcinol concentration, the kinetic data follows the Langmuir-Hinshelwood first-order kinetics, which is expressed as:ln (C_o_/C_t_) = kt(1)
where C_o_ (mM) is the resorcinol initial concentration, C_t_ (mM) is the concentration at time t, and k is the apparent rate constant. It can be seen in Figure 8 that the plot of ln (C_o_/C_t_) against time t gives straight lines with the four different catalyst amounts of 50, 100 m, 150, and 200 mg. An increase in the value of k, and hence the rate constant, was observed with an increase in the catalyst concentration from 50 to 150 mg.

This increase can be described by the presence of a larger number of available active sites on the catalyst surface, which promote the effective photocatalytic degradation. However, at higher amounts of catalyst, i.e., 200 mg, the rate constant value decreases, possibly due to the aggregation of the catalyst in the reaction medium. Thus, the amount of the catalyst is a crucial parameter for the photocatalytic removal of pollutants from the aqueous environment [49,50]. The maximum value of k was found to be 4.65 × 10^−3^ s^−1^ with 150 mg of catalyst.

### 2.13. Antibacterial Application

The prepared rGO-ZnS and 3% rGO-ZnS-Ag nanocomposite was also evaluated in the destruction of the harmful bacterial strains, i.e., *E. coli* (Gram-negative G). To evaluate the antimicrobial activity, the mean inhibition zone was measured in the disk diffusion test (Figure 9).

With the rGO-ZnS-Ag nanocomposite, the inhibition zone of 18 mm for 2 mg/mL, 15 mm for 1 mg/mL, 11 mm for 0.5 mg/mL, and 06 mm for 0.25 mg/mL were measured against *E. coli*. Similarly, with the rGO-ZnS nanocomposite, the inhibition zone of 16.5 mm for 2 mg/mL, 13.5 mm for 1 mg/mL, 09 mm for 0.5 mg/mL, and 05 mm for 0.25 mg/mL were measured. As can be observed from the above-mentioned inhibition zones, increasing the concentration of the rGO-ZnS and rGO-ZnS-Ag nanocomposite, the effectiveness against harmful pathogens also increases. This is also consistent with previously reported results [51,52]. Thus, both the rGO-ZnS and rGO-ZnS-Ag nanocomposites could effectively destroy harmful pathogens in polluted water and have potential application in waste water treatment, just like other noble metal/graphene oxide nanocomposites.

## 3. Experimental Part

### 3.1. Materials

Graphite powder (99.9%), silver nitrate (NaNO_3_, 99.5%), zinc acetate (II) dehydrated (Zn (CH_3_COO)_2_.2H_2_O, 99.9%), sodium sulphide (Na_2_S, 99.99%), and N, N-Dimethyl formamide (N(CH_3_)_2_CHO) were bought from sigma-aldrich. Malachite green (MG, 90%) and ethyl violet (EV, 90%) dyes were supplied by Aldrich chemicals. Resorcinol (C₆H₆O_2_, 99.5%) was supplied by Scharlau. All the chemicals were used without further purification. Additionally, double-distilled water (DDW) was utilized throughout the experiments for the cleaning of apparatuses and solution preparation.

### 3.2. Graphene Oxide Synthesis

Graphite powder was used as the starting material to prepare GO by using modified Hummer’s method, as cited in the previous literature [7,9]. In this method, graphite powder (1 g) and sodium nitrate (NaNO_3_, 1 g) were mixed under an ice bath (0−4 °C) in 25 mL sulfuric acid (H_2_SO_4_) at constant stirring. After 1.5 h stirring, 3 g of KMnO_4_ were added to the mixture very gently so that the temperature may not rise above 15 °C. Next, the mixture was diluted with addition of 100 mL of distilled water and kept on stirring for 2 h. Then, we removed the ice bath to raise the temperature to 35 °C. The reaction mixture was refluxed at 98 °C for 15 min with the addition of 100 mL of distilled water. Finally, the solution was treated with 30% H_2_O_2_ and the color appeared bright yellow, indicating the formation of GO. This GO suspension, after washing with distilled water at least five times to remove unreacted species, was vacuum dried for the preparation of the nanocomposite.

### 3.3. Reduced Graphene Oxide-ZnS Nanocomposite Synthesis

For rGO-ZnS nanocomposite preparation, firstly 0.05 M zinc acetate solution was prepared in DMF solvent under 10 min vigorous stirring. Then, 10 mL of 0.6 mg/mL of freshly prepared graphene oxide suspension was added to zinc acetate solution. Solution was ultrasonicated for about 15 min. Next, 0.05 M freshly prepared sodium sulfide solution was added slowly under continuous magnetic stirring. The resulting mixture was then allowed to stir for another two-and-a-half hours. DMF not only act as a solvent, but also a reducing agent for both Zn^+2^ ion and graphene oxide. Greyish suspension indicated the formation of rGO-ZnS nanocomposite, which was then centrifuged, washed with distilled water, and dried at 60 °C in an oven for further use.

### 3.4. Synthesis of SILVER (Ag) Doped rGO-ZnS Nanocomposite

Zinc Acetate (ZnC₄H₆O₄) 1.04 g and silver nitrate (AgNO₃) 0.32 g were added to 60 mL of DMF solution. Then, 20 mL of 0.6 mg/mL of freshly prepared graphene oxide suspension was added to the above DMF solution. Na₂S solution was added drop wise to the above-mentioned DMF solution, and the colour of the solution turned dark brown. This doped composite solution was then centrifuged and washed with distilled water many times and dried in an oven for about 12 h. After completely drying, the doped nanocomposite material was named as rGO-ZnS-Ag.

### 3.5. Adsorption Experiment

For adsorption studies of dyes on rGO-ZnS nanocomposite, the batch method was used to study in detail the effect of different factors, such as pH, dyes initial concentrations, and initial adsorbent dose. Aqueous solutions of 4 different starting concentrations of 3, 5, 7, and 10 μM of both dyes were prepared. UV-Vis spectrophotometer was used as a monitoring tool for detailed adsorption kinetics of dyes onto nanocomposite. A total of 5 μM (50 mL) solution of both dyes were taken in separate conical flasks, along with an equal amount of nanocomposite (3 mg), and the solutions were stirred at a speed of 100 rpm. About 2 mL of reaction mixture was taken out from the flask at intervals of 5 min, until 1 h, and the adsorbed amount was calculated using the following equation:q_t_ = (C_o_ − C_t_) V/m (2)

In the above equation, concentrations (ppm) of dyes at zero-time (C_o_) and at time t (C_t_) were measured using the UV–Visible spectrophotometer at their maximum absorption wavelength of 617 nm and 595 nm for MG and EV, respectively. V and m refer to the volume of the solution and the mass of the adsorbent. The pH effect on the dyes’ adsorption was also studied, using 5 μM aqueous solutions of both dyes and using 0.1 M HCl and 0.1 M NaOH aqueous solutions for pH adjustment. The dyes’ initial concentration effect was studied with 4 different concentrations (3, 5, 7, and 10 μM) for each dye, using 3 mg of adsorbent. Similarly, to study the adsorbent dose effect, 3 different amounts of the adsorbent (3, 5, and 7 mg) were taken separately with the fixed initial concentration of each dye (5 μM), and the adsorbed amount q_t_ (in mg/g) was calculated using the above Equation (2). Obtained adsorption data were also treated with various adsorption isotherms, along with kinetics models.

### 3.6. Photo-Degradation Experiment

The photodegradation experiment of the prepared rGO-ZnS-Ag nanocomposites was performed for the degradation of resorcinol from polluted water under direct sunlight. For this, 20 ppm, 1000 mL of resorcinol aqueous solution was prepared in a 1000 mL volumetric flask. For the adsorption study, firstly an adsorption equilibrium was established in the dark without UV light source, in which 10 mg of rGO-ZnS-Ag was added to 100 mL of prepared resorcinol aqueous solution and kept under magnetic stirring for about 1 h to accomplish the adsorption/desorption equilibrium. The adsorbed concentration of resorcinol on synthesized photocatalysts was then determined by UV-Vis spectrophotometer at a time interval of 30 min. At the completion of the adsorption process, the same test solutions of resorcinol were irradiated under sunlight, and the photodegradation kinetics was studied with a specific time lag of about 30 min for the resorcinol aqueous solution. For this, 2 mL of the irradiated test solution of the pollutant was taken out at a given interval and centrifuged for 10 min at 5000 rpm for the separation of the photocatalyst from the aqueous solution. The absorbance of the centrifuged resorcinol solution was then determined spectrophotometrically at a λ_max_ of 274 nm for the resorcinol.

### 3.7. Sample Preparation for Antimicrobial Studies

Antibacterial studies of the prepared rGO-ZnS and rGO-ZnS-Ag nanocomposite were tested against the bacterial strain, *E. coli*. The zone of inhibition was studied with Moller Hinton agar (MHA). From pure culture, morphologically comparable colonies of bacterial culture were transferred and bacteria were full-grown aerobically for about 18 h at 37 °C. Nutrient agar was dispersed in 25 mm disinfected Petri dishes. These disinfected plates were stored in sealed plastic bags at 4–8 °C. Mueller Hinton agar media was prepared by dispersing Mueller Hinton agar (38 g/L) in distilled water medium and autoclaved at 121 °C and 15 psi for about 20 min. The plates were seeded with inoculum (bacterial culture in MHA) using sterile cotton swabs. Five wells in each plate were made by using sterile steel borer (8 mm). Then, 50 μL of the composite (prepared in autoclave distilled water) with 2 mg/mL, 1 mg/mL, 0.5 mg/mL, 0.25 mg/mL were dispensed in respective wells labelled as 1, 2, 3, and 4, respectively. The incubation period was 24 h at 37 °C.

### 3.8. Characterization Techniques for Prepared Samples

Synthesized nanocomposite was characterized as follows. To have a look at the crystallinity and purity of the prepared sample, XRD was performed with Shimadzou, Kyoto, Japan XRD-6000 (Cu Kα radiation λ = 0.154 nm) in a range of 2θ from 5° to 80° at a scan rate of 8° min^−1^. For morphological studies of synthesized nanomaterial, FE-SEM image was recorded with the FESEM-Hitachi S4800 scanning electron microscope. TEM Hitachi Tecnai G20, Tokyo, Japan was used to examine minor structural details of the nanocomposite in synergy with SEM results. For TEM analysis, the sample was first sonicated in methanol for 30 min and then a diluted drop of that sample was placed on a carbon grid and dried overnight. To have a look at the elemental composition, EDX was carried out, which is attached to the FE-SEM to gain insight on the number and types of elements present. Thermogravimetric analysis called TGA was carried out on a Mettler toledo, Greifensee, Switzerland, thermogravimetric analyser under nitrogen environment and at a heating rate of 10 °C min^−1^. For monitoring the optical property of the nanocomposite, UV-Vis spectra were recorded by UV-Vis spectrophotometer (Shimadzu, Kyoto, Japan), and also to study the adsorption experiments of the dyes and photocatalytic reactions of the phenolic compound. Surface functionalities of prepared nanocomposites and dye-loaded nanocomposite were recorded using a FT-IR spectrometer [Brooker tensor II, Ettlingen, Germany] in a range of 4000–600 cm^−1^. The surface area, pore size, and pore volume were analyzed from nitrogen gas adsorption/desorption studies performed at 77.35 K. The outgas time and temperature were kept as 8 h and 393 K, respectively. The corresponding parameters were calculated via the single-point and multi-point Brunauer–Emmett–Teller (BET); Langmuir; Density Functional Theory (DFT); Barrett, Joyner, and Halenda (BJH); Dollimore; and Heal (DH) methods.

## 4. Conclusions

From the present study, it is concluded that the rGO–ZnS and rGO-ZnS-Ag nanocomposites, which were prepared by a facile one-step synthesis process, have been found to work as efficient adsorbent materials for the removal of the hazardous industrial dyes MB and EV from an aqueous environment, as an efficient photocatalyst system for the degradation of the aromatic pollutant resorcinol, and as an antibacterial agent. Various fundamental parameters, such as the dyes’ initial concentrations, the pH of solution, and the interaction time, have been found to tune the maximum adsorption capacity for the rGO-ZnS nanocomposite. Further, adsorption kinetics concluded that the adsorption data of both dyes fitted well to the pseudo-second order kinetic model, with the values of q_e_ found to be 27.54 mg/g and 20.04 mg/g for MG and EV, respectively, which matches well with the experimental values of 25.72 mg/g and 18.34 mg/g for both dyes. Additionally, rGO-ZnS showed good recyclability for up to four consecutive cycles, indicating an efficient adsorbent material for industrial purposes. Furthermore, doped nanomaterial, i.e., the rGO-ZnS-Ag nanocomposite, showed an excellent photocatalytic potential in the degradation of a representative phenolic compound, resorcinol. Parameters such as the pH, initial concentration of resorcinol, and photocatalyst loading, have been optimized to obtain a maximum degradation efficiency, and the kinetics of degradation have also been studied in detail. Maximum degradation was found at pH 7, with an optimum photocatalyst dose of 150 mg. Another advantage of the nanocomposites under study was their good antibacterial activity against harmful bacterial pathogens, with an inhibition zone of 18 mm against a 2 mg/mL concentration of rGO-ZnS-Ag. Thus, Ag-decorated rGO-ZnS nanosheets act as an excellent photocatalyst for the degradation of phenolic compounds. Therefore, the present study might open up new opportunities for the use of semiconductor-based materials, not only as efficient photocatalysts for the degradation of various organic contaminants, but also as efficient adsorbents and antimicrobial material for waste water treatment.

## Figures and Tables

**Figure 1 molecules-28-00926-f001:**
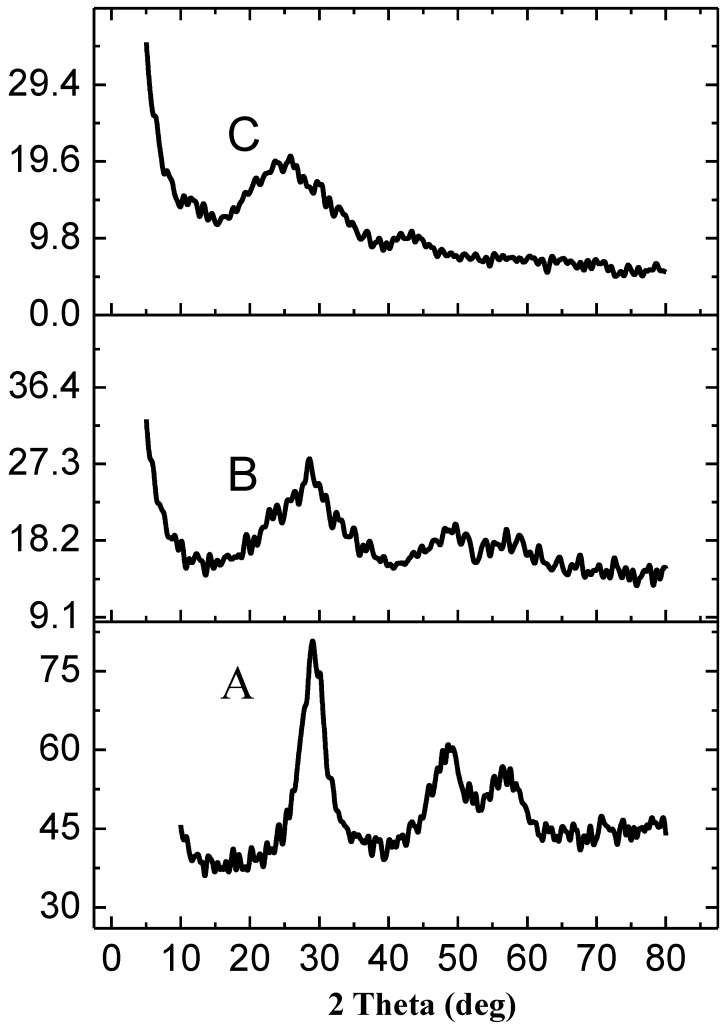
XRD pattern of rGO-ZnS (**A**), rGO-ZnS-Ag (**B**), and rGO (**C**).

**Figure 2 molecules-28-00926-f002:**
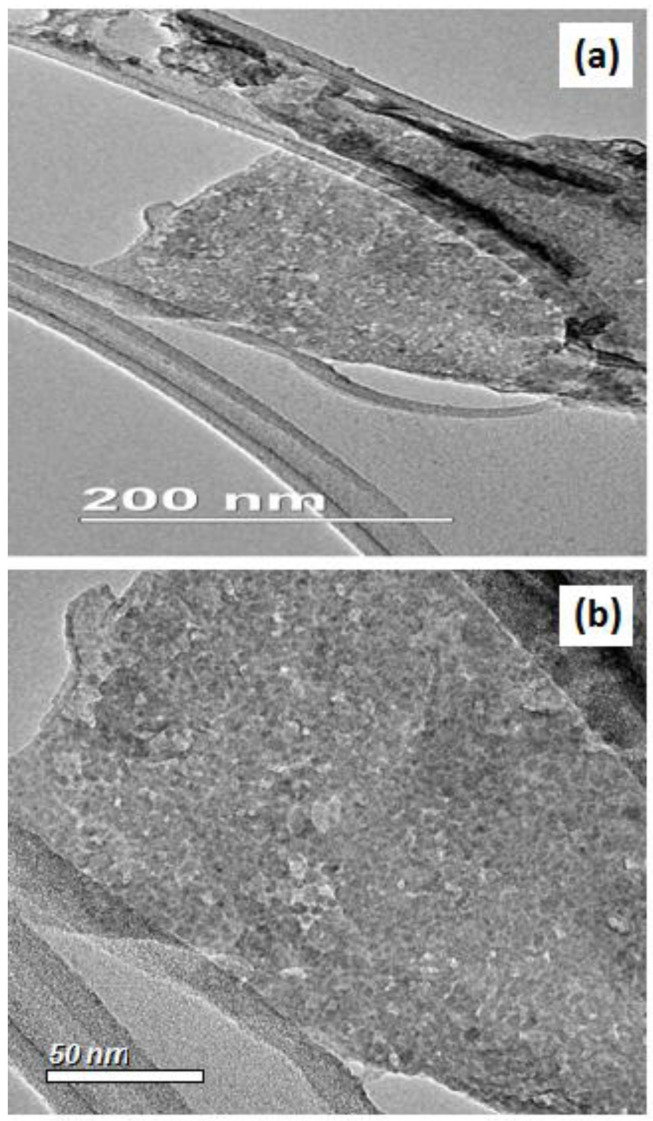
TEM images of rGO-ZnS at (**a**) low and (**b**) high magnification.

**Figure 3 molecules-28-00926-f003:**
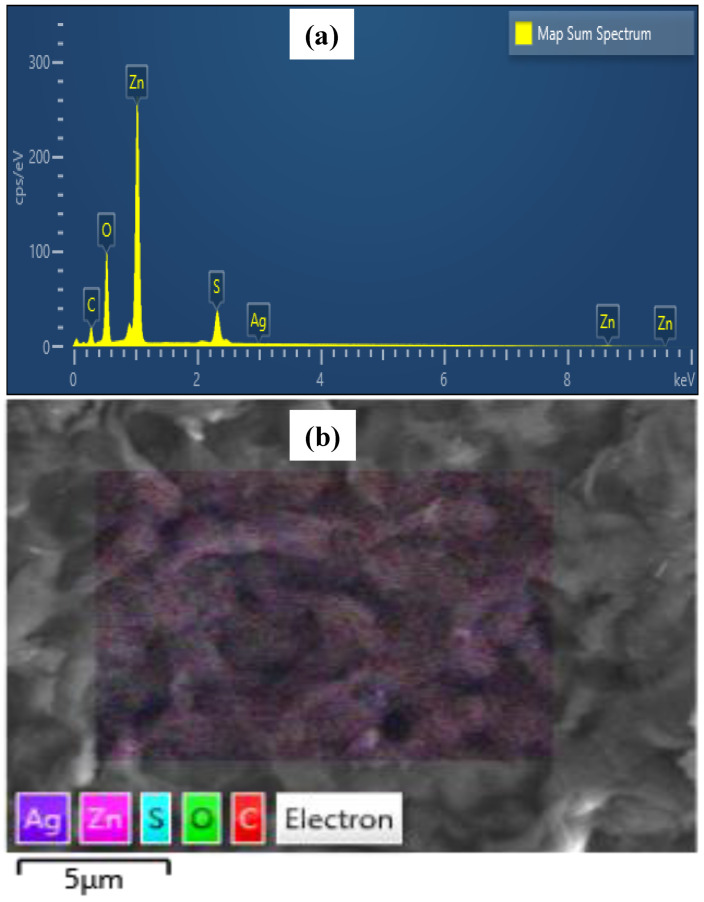
EDX spectrum (**a**) and SEM-EDX elemental mapping (**b**) of rGO-ZnS-Ag.

**Figure 4 molecules-28-00926-f004:**
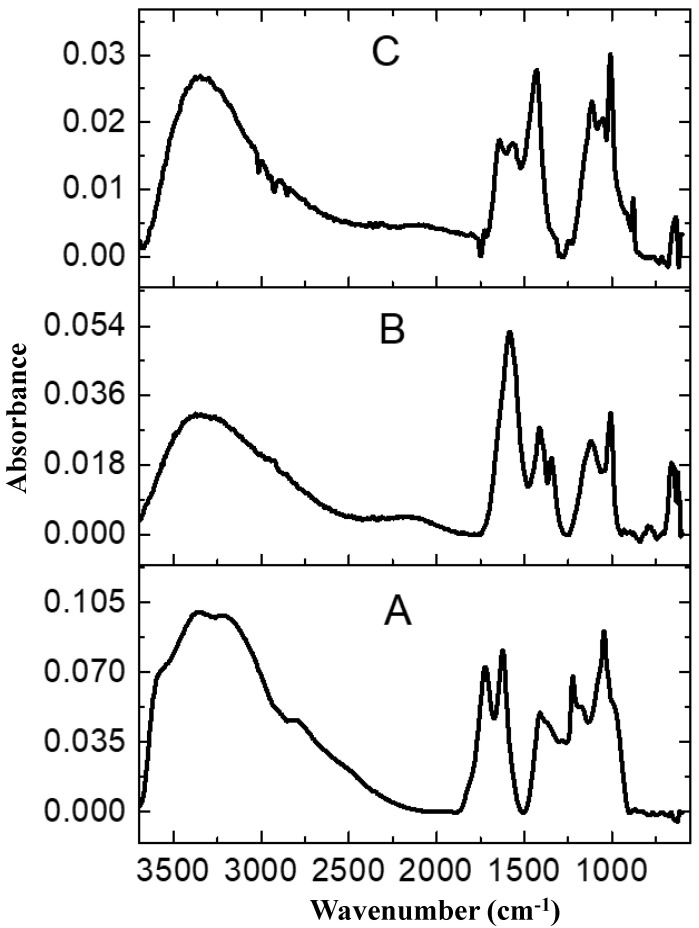
FTIR spectra of rGO (**A**), rGO-ZnS (**B**), rGO-ZnS-Ag (**C**).

**Figure 5 molecules-28-00926-f005:**
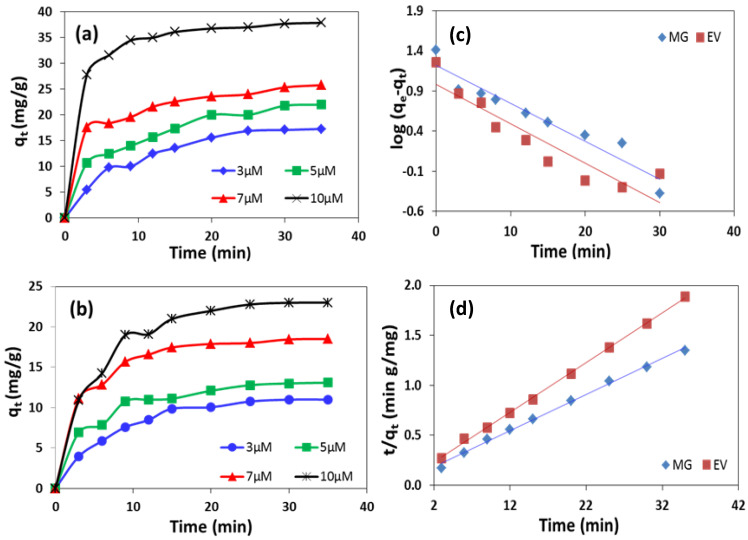
Effect of initial concentration of dyes on the adsorbed amount of (**a**) MG and (**b**) EV on rGO-ZnS nanocomposite at 25 °C. Plots of (**c**) pseudo first-order model and (**d**) pseudo second-order model for MG and EV adsorption.

**Figure 6 molecules-28-00926-f006:**
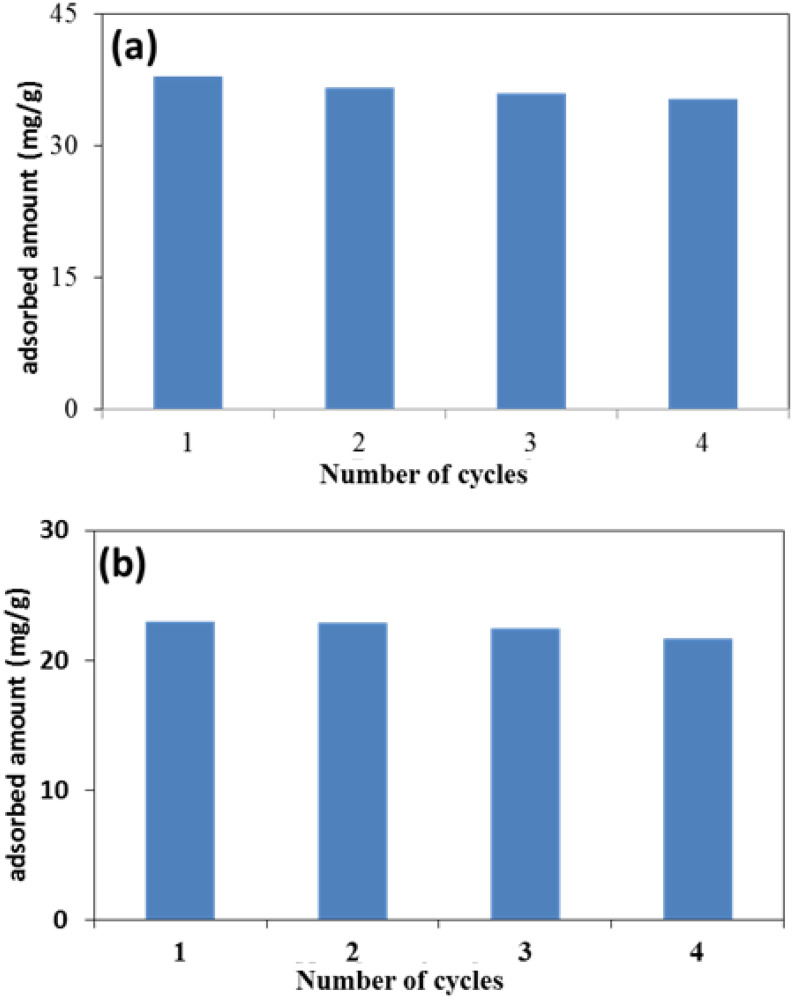
Desorption efficiency plots of (**a**) MG in 0.1 M HCl, (**b**) EV in ethanol solvent up to 4 successive cycles.

**Figure 7 molecules-28-00926-f007:**
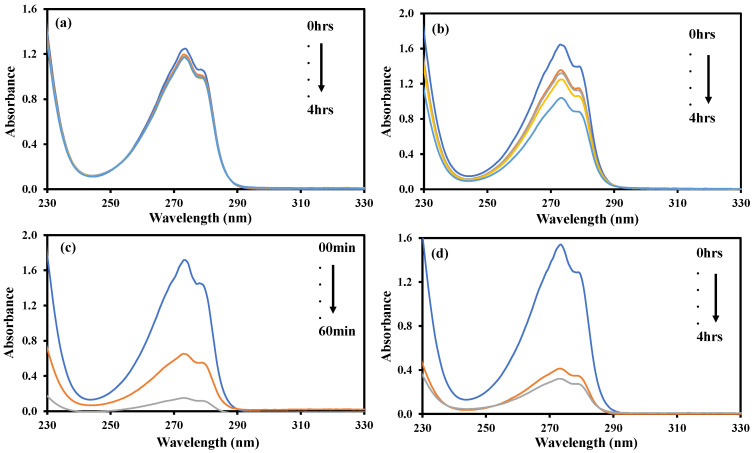
Rate of resorcinol degradation at pH values of (**a**) 2, (**b**) 4, (**c**) 7, (**d**) 9.

**Figure 8 molecules-28-00926-f008:**
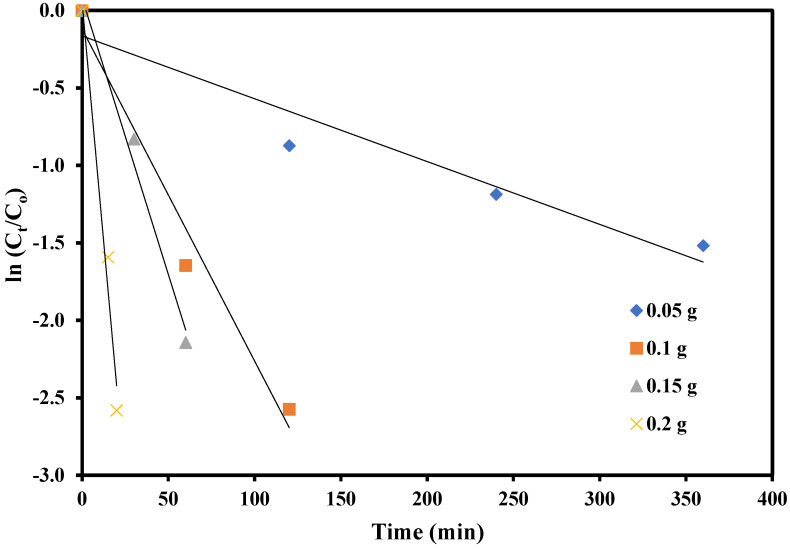
Rate constant values at different amounts of catalyst for the reduction of resorcinol using GO-Ag-ZnS nanocomposite as catalyst.

**Figure 9 molecules-28-00926-f009:**
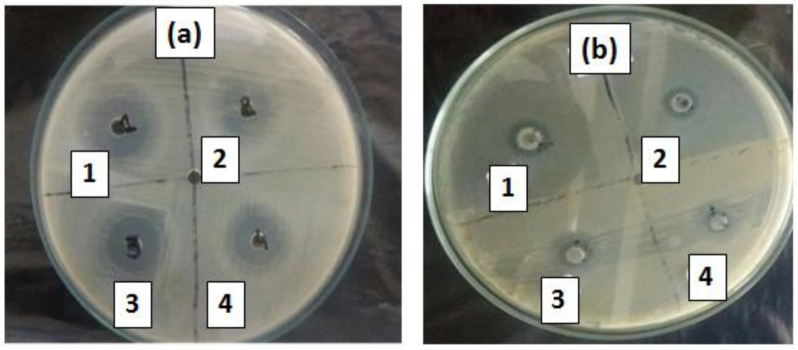
Usage of (**a**) rGO-ZnS-Ag, (**b**) rGO-ZnS nanocomposite against Eschericia coli. [(1) 2 mg/mL, (2) 1 mg/mL, (3) 0.5 mg/mL, and (4) 0.25 mg/mL.].

**Table 1 molecules-28-00926-t001:** Surface area, pore volume, and pore radius of rGO-ZnS-Ag nanocomposite.

Method	Single Point BET	Multi Point BET	BJH Cumulative Adsorption	BJH Cumulative Desorption	DH Cumulative Adsorption	DH Cumulative Desorption	Langmuir	DFT Cumulative
Surface Area (m^2^/g)	456.1	550.3	431.6	457.2	441.9	468.3	1082	291.2
Pore volume (cc/g)	-	-	0.6898	0.6991	0.6791	0.6888	-	0.6892
Pore Radius (Å)			16.21	16.02	16.21	16.02		15.85

**Table 2 molecules-28-00926-t002:** Kinetic parameters for the adsorption of MG and EV on to the rGO-ZnS nanocomposite.

Kinetics	Constants	Dyes	
		MG	EV
Pseudo-first order model	k_1_ (min^−1^)	0.108	0.11
	q_e_ (mg/g)	16.5	9.63
	R^2^	0.9242	0.8543
Pseudo-second order Model	k_2_ (g/mg min)	0.011	0.019
	q_e_ (mg/g)	27.54	20.04
	R^2^	0.9961	0.999

## Data Availability

The data presented in this study are available on request from the corresponding author. The data are not publicly available due to privacy.

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
