# Peer review of "Reduced Graphene Oxide-Zinc Sulfide Nanocomposite Decorated with Silver Nanoparticles for Wastewater Treatment by Adsorption, Photocatalysis and Antimicrobial Action"

_molecules, 2023, doi:10.3390/molecules28030926_

Round 1
Reviewer 1 Report
Wastewater treatment is important for the sustainable development. rGO and its nanocomposites are promising materials for wastewater treatment attributing to their high specific surface area. The research of this manuscript is interesting and results are reliable. However, major revision is required and the comments are given below.
1. The introduction is suggested to be divided into several paragraphs.
2. Wastewater treatment is a hot topic and many absorbents with good performance are developed. It would be better to describe different absorbents in the introduction part for broad readers. More references are suggested to be cited, especially those newly published. Please refer and cite “A review on conversion of crayfish-shell derivatives to functional materials and their environmental applications; Synthesis and Application of Granular Activated Carbon from Biomass Waste Materials for Water Treatment: A Review; MOFs meet wood: Reusable magnetic hydrophilic composites toward efficient water treatment with super-high dye adsorption capacity at high dye concentration”.
3. Please revise “silver nitrate (NaNO3, 99.5%)” and double check the whole manuscript to remove such typos.
4. A full stop is missing for the sentence “Resorcinol (C₆H₆O2, 99.5%) was supplied by Scharlau”.
5. There should be a space between the number and the unit. Please go through the whole manuscript to revise the writing of units.
6. “This GO suspension, after drying, was used further for preparation of nanocomposite.” The GO suspension should be purified by dialysis to remove salts and byproducts before drying. The drying method of GO suspension should also be stated.
7. “Zn+2 ion” need to be revised as “Zn2+ ions”.
8. Why the XRD patterns are so noisy? Y axial is no need to be labeled.
9. SEM images in Figure 2 looks wired. It seems like cut and paste. Please insert original images to keep high resolution.
10. Most of the references are too old. Please added more references published in recent years.
Reviewer 2 Report
The comments are as follows.The organization of the manuscript is poor with up to 14 figures presented in the main text. In fact, most of the figures are either not critical (can be arranged in supplementary files) or not required.
The title is about wastewater. But throughout the manuscript, authors only used very simple synthetic solution for testing (either contain 1 dye or resorcinol). The wastewater is far more complicated compared to the solutions tested in this work.
Furthermore, I really can’t find the relationship between photocatalysis and antimicrobial action. The research gap of the work is NOT clear at all. I have no idea what the key problems the authors intended to address. The synthesized nanomaterials as reported in this work have been previously reported elsewhere. Authors also failed to provide relevant review on the subject topics (In Introduction) and what is the current progress and what they wanted to solve, etc.
The current Introduction only contains 1 long paragraph with no critical information related to the subject topic and relevant prior art.
Figure 3 is about TEM image, but the authors still supplied much lower image resolution in Figure 2 (SEM). We can’t draw any conclusion on the nanomaterials examined under SEM. Furthermore, 3 nanomaterials were synthesized, but the authors only selectively chose 1 type for SEM/TEM analysis.
Other key properties of the nanomaterials, i.e., particle size and BET surface area are missing. They are critical for adsorption and photocatalysis process.
Figure 5 – Most of the important peaks are not discussed. For instance, the broad peak at 3200 cm-1 as found in all samples but at different intensity is not discussed. All these spectrum should be arranged on same image for ease comparison.
Figure 6-7 – The presentation of the figures is NOT consistent. Besides, most of the key findings are not compared with relevant studies.
Figure 9 – The desorption studies should be carried out on resorcinol – the key pollutant identified in this work.
Figure 11 – What is the adsorption of nanomaterials towards resorcinol prior to photocatalysis?
Figure 14 – How can the results be related to photocatalysis?
Round 2
Reviewer 1 Report
The manuscript is acceptable now. One more suggestion is to fix reference 8. Some information for reference 8 is missing. The author is suggested to replace reference 8 with a newly published reference "Gotore Obey; Munodawafa Adelaide; Rameshprabu Ramaraj, Biochar derived from non-customized matamba fruit shell as an adsorbent for wastewater treatment. Journal of Bioresources and Bioproducts 2022, 7 (2), 109-115."
Reviewer 2 Report
It is completely unacceptable to add two additional authors (Abdul Hai and Syeda Huma H. Zaidi) at this stage. I don't see the need to have two additional authors for the revised manuscript. The comments given in the first round of review are not critical and the authors can easily revise it without having two additional authors. I strongly against the unethical practise of the corresponding author.
Round 3
Reviewer 2 Report
no comment